# Quantum critical phase of FeO spans conditions of Earth's lower mantle

Wai-Ga D. Ho[1,6], Peng Zhang[2,6] ✉, Kristjan Haule [3], Jennifer M. Jackson[4], Vladimir Dobrosavljević [1] & Vasilije V. Dobrosavljevic [4,5] ✉

Seismic and mineralogical studies have suggested regions at Earth's core-mantle boundary may be highly enriched in FeO, reported to exhibit metallic behavior at extreme pressure-temperature ($P–T$) conditions. However, underlying electronic processes in FeO remain poorly understood. Here we explore the electronic structure of $B1$-FeO at extreme conditions with large-scale theoretical modeling using state-of-the-art embedded dynamical mean field theory (eDMFT). Fine sampling of the phase diagram reveals that, instead of sharp metallization, compression of FeO at high temperatures induces a gradual orbitally selective insulator-metal transition. Specifically, at $P–T$ conditions of the lower mantle, FeO exists in an intermediate quantum critical state, characteristic of strongly correlated electronic matter. Transport in this regime, distinct from insulating or metallic behavior, is marked by incoherent diffusion of electrons in the conducting $t_{2g}$ orbital and a band gap in the $e_g$ orbital, resulting in moderate electrical conductivity (~$10^5$ S/m) with modest $P–T$ dependence as observed in experiments. Enrichment of solid FeO can thus provide a unifying explanation for independent observations of low seismic velocities and elevated electrical conductivities in heterogeneities at Earth's mantle base.

Earth's lower mantle is thought to be composed primarily of bridgmanite ($Mg_{1-x}Fe_x)SiO_3$ and ferropericlase $(Mg_{1-x}Fe_x)O$, where $x$ ~0.1–0.2, coexisting with $CaSiO_3$[1–3]. These major mineral phases behave as insulating materials up to conditions of the lowermost mantle, with electrical conductivities on the order of $10^0$ to $10^2$ S/m[4,5], many orders of magnitude lower than proposed conductivities of the metallic iron-dominant core (~$10^6$ S/m) (e.g., refs. [6,7]). Instead of a homogeneous lower mantle, seismic observations over the last several decades have robustly identified multi-scale structures across Earth's core-mantle boundary[8,9]. These structures have been grouped into two main categories: (1) two continent-scale "large low-seismic velocity provinces" (LLSVPs), considered to be piles of heterogeneous material or bundles of thermochemically distinct mantle plumes[10,11], and (2) numerous mountain-scale "ultralow velocity zones", basal structures discovered within and around the edges of LLSVPs, including at the roots of major mantle plumes like those that source volcanism at Hawai'i, Iceland, and the Gálapagos[12–19].

Studies generally agree that the interpretation of these observed structures requires strong compositional contrasts from the surrounding average lower mantle and possibly the presence of partial melt[14,20,21]. Recent interdisciplinary work on ultralow velocity zones has demonstrated that solid FeO-rich mineral assemblages, consisting of iron-rich $(Mg_{1-x}Fe_x)O$ ($x$ ~0.8–0.95) coexisting with $(Mg,Fe)SiO_3$ and $CaSiO_3$, can produce structures that satisfy the velocity reductions and

[1]Department of Physics and National High Magnetic Field Laboratory, Florida State University, Tallahassee, FL, USA. [2]MOE Key Laboratory for Non-equilibrium Synthesis and Modulation of Condensed Matter, Shaanxi Province Key Laboratory of Advanced Functional Materials and Mesoscopic Physics, School of Physics, Xi'an Jiaotong University, 710049 Xi'an, Shaanxi, PR China. [3]Center for Materials Theory, Department of Physics, Rutgers University, Piscataway, NJ, USA. [4]Seismological Laboratory, California Institute of Technology, Pasadena, CA, USA. [5]Present address: Earth and Planets Laboratory, Carnegie Institution for Science, Washington, DC, USA. [6]These authors contributed equally: Wai-Ga D. Ho, Peng Zhang. ✉e-mail: zpantz@mail.xjtu.edu.cn; vasilije@carnegiescience.edu

topographies constrained by seismic observations and geodynamic simulations[22–26]. Such strong iron enrichment, arising from crystallization of the primordial magma ocean or chemical interactions with the iron core, leads to several unique physical properties observed for the very iron-rich (Mg,Fe)O phase, including high seismic anisotropy[27], remarkably low viscosity[28], and experimental reports of moderately elevated electrical conductivity ($10^5$ to $10^6$ S/m)[29,30], orders of magnitude higher than insulators (like typical mantle rocks) but lower than a metal (like the liquid iron-rich core).

The accuracy and origin of these intermediate conductivity values, and the electronic phase diagrams of FeO and iron-rich (Mg,Fe)O more broadly, represent a poorly understood and controversial topic in high-pressure physics and deep Earth science. An insulator-metal transition has been proposed for FeO from measurements of relatively high conductivity (~$10^5$ S/m) with weak $P–T$ dependence above ~60 GPa[29]. In contrast, similarly high conductivity was reported for $(Mg_{0.2}Fe_{0.8})O$ and $(Mg_{0.05}Fe_{0.95})O$ but interpreted as insulating behavior up to ~130 GPa[30]. Meanwhile, standard electronic-structure theory methods focus at $T = 0$ K, and are not able to properly capture thermal effects, which often dominate in the vicinity of the insulator-metal transition[31,32]. Such extreme fragility of electronic states is especially pronounced in "strongly correlated"[33] electronic systems[34–36], often featuring tightly-bound $d$ or $f$ orbitals[37]. Here the Coulomb repulsion between pairs of electrons confined to the same orbital takes center stage, typically resulting in very strong electron-electron scattering and poor conduction at elevated temperature[38]. Given these complications, several fundamental open questions arise regarding the insulator-metal transition (IMT) in $B$1-FeO at high pressures: (1) Is there a sharp IMT at high temperature, in the regime characteristic of Earth's deep mantle? (2) What is the mechanism of electronic transport (i.e., the dominant form of scattering) in this regime? (3) How do orbital selectivity[39] and the associated spin-crossover affect the transition region? (4) What are the consequences of these phenomena for the magnitude and $P–T$ dependence of electrical conductivity across deep Earth conditions?

Knowledge of electronic processes in FeO at extreme conditions and consequences for transport properties is essential for understanding phenomena at Earth's core-mantle boundary, including electromagnetic coupling of the core and mantle and heat flow through this region. To that end, we employ a state-of-the-art "embedded DMFT" (eDMFT) ab initio approach[40] that combines dynamical mean field theory (DMFT) methods[41,42] and standard density functional theory (DFT) with full charge self-consistency. While some valuable steps in this direction have been taken in previous work[29,43–45], sufficiently detailed and systematic study of the transition region has not been performed, preventing a clear understanding of the important open questions at hand. Using this approach, we systematically survey the electronic structure of cubic $B$1-FeO, the crystal structure relevant to Earth's lower mantle conditions[46]. An expansive data set featuring calculations at more than 350 temperature-volume conditions (see Supplementary Materials) finely samples the phase diagram up to conditions of Earth's inner core (300 GPa, 5000 K). This detailed information allows us to accurately determine and physically interpret the boundaries of different transport regimes across the phase diagram.

## Results and discussion
### Three distinct electronic phases of $B$1-FeO
Our theoretical calculations reveal three distinct electronic phases in the high-$P–T$ phase diagram of $B$1-FeO (Fig. 1). At ambient conditions and low degrees of compression, FeO behaves as a Mott insulator, in which both the $t_{2g}$ and $e_g$ orbitals exhibit large band gaps at the Fermi energy on the order of several eV and electrons remain bound to their respective nuclei[29,45]. In contrast, at large degrees of compression, FeO exists as a strongly correlated metal, where one or both the $d$ orbital

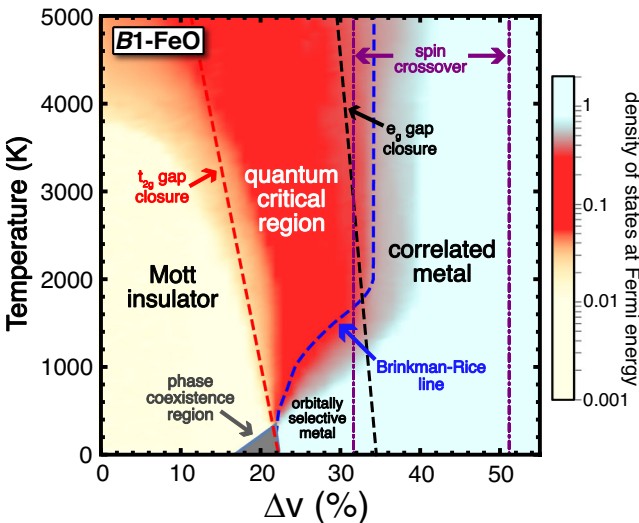

**Fig. 1 | Theoretical phase diagram of $B$1-FeO, as a function of reduced volume Δv and temperature $T$.** Δv = $(v_o − v)/v_o$, where $v_o$ is the volume of FeO at ambient conditions. The color-coded value of the electronic density of states (DOS) at the Fermi energy is used to distinguish the (gapped) Mott insulator from the metal and the intermediate "quantum critical" regime[34,49]. The "Brinkman-Rice" crossover line[82] marks the thermal destruction of coherent quasiparticles in the metal with increasing temperature[83].

band gaps are closed, producing a characteristic "quasiparticle" density of states (DOS) peak at the Fermi energy (see also Fig. 2, rightmost panels)[29,45].

At intermediate degrees of compression and sufficiently high temperatures, FeO exists in a "quantum critical" (QC) state, which is notably different from either an insulator or a metal. Here, the $t_{2g}$ gap has closed to form a conducting band, but unlike in a conventional metal, the density of states at the Fermi energy is significantly reduced, with a marked absence of quasiparticles (Fig. 2, bottom row). Instead of traveling as coherent waves with minimal scattering as in a metal, electrons in the QC state exhibit incoherent diffusion marked by strong electron-electron scattering with a short mean-free path at the scale of atomic spacing. In this regime, the $e_g$ gap remains open and FeO remains in the high-spin state, with four $d$ electrons in the $t_{2g}$ orbital (Fig. 3). We stress that the QC phase arises only at finite temperatures above the insulator-metal phase coexistence region, terminating at the critical end-point $T_c$ ~370 K; the insulator-metal transition assumes first-order character at $T < T_c$.

### Temperature-dependent forms of the IMT
The physical nature of the insulator-metal transition in FeO and the range of pressures spanning the QC region depend strongly on the range of temperatures considered. At low temperatures ($T \lesssim T_c$), FeO transitions directly from a Mott insulator to an "orbitally selective" metal around Δ$v$ ~20% (corresponding to $P$ ~58 GPa[46]). Here the closure of the $t_{2g}$ gap leads to the immediate formation of a quasiparticle peak at the Fermi energy in the $t_{2g}$ orbital (see Fig. 2, top row), while the $e_g$ gap remains open. These quasiparticle states are remarkably fragile to thermal excitations, and are suppressed around the "Brinkman-Rice" temperature $T_{BR}$ (see Fig. 1), marking the crossover to the QC phase. As $T_{BR}$ increases with compression, the insulator-metal transition is "smeared out", producing an increasingly wider QC "fan" at $T_c < T \lesssim 2000$ K. The left boundary of the QC region corresponds to a temperature scale where the Mott gap is smeared through thermally activated processes (see Supplementary Materials for precise definition of the corresponding crossover lines shown in Fig. 1).

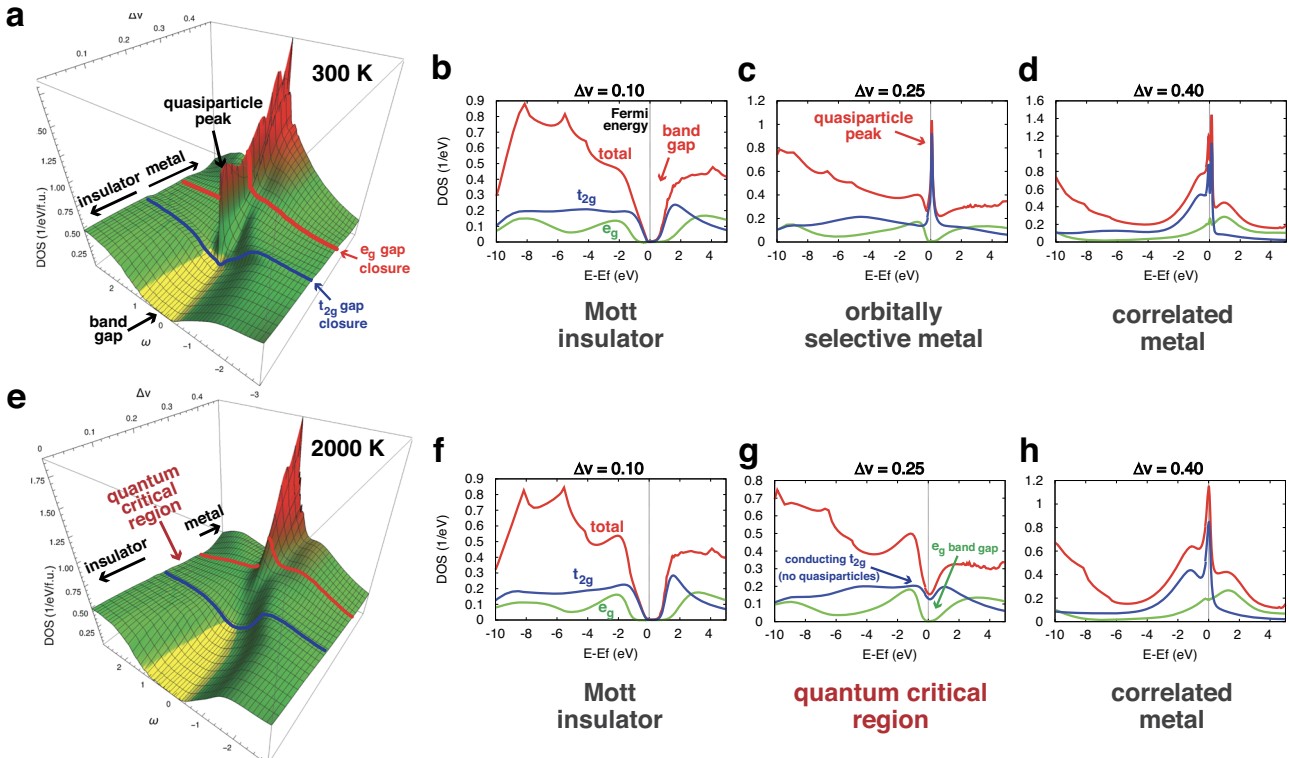

**Fig. 2 | Evolution of the electronic density of states (DOS) with compression.**
**a–d** Metallization is sharp at low temperature ($T = 300$ K), where closure of the Mott gap in the $t_{2g}$ band leads to immediate emergence of coherent quasiparticle states at the Fermi energy. **e–h** A broad intermediate "quantum critical" phase arises at higher temperatures ($T = 2000$ K), with a spectral pseudogap (reduced but finite DOS) and no quasiparticle states, characteristic of an incoherent conductor. The quasiparticle peak appears only at further compression with the closure of the $e_g$ gap and onset of the spin crossover phenomenon.

This behavior becomes qualitatively different at very high temperatures. At $T \gtrsim 2000$ K, the quasiparticles are unable to form in the $t_{2g}$ orbital before compression causes the closure of the $e_g$ gap, around $\Delta v \sim 34\%$. Further compression leads to the onset of spin crossover phenomena and simultaneous formation of a correlated metal, with robust quasiparticles forming in both sectors. The spin crossover extends over a wide compression range with weak temperature dependence (Fig. 3) as previously observed[47], and is marked by a partial charge transfer from the $e_g$ to the $t_{2g}$ orbital, with one electron remaining in the $e_g$ orbital and a drop in the magnetic moment from 4 to ~1.5 Bohr magneton. Unlike $T \lesssim 2000$ K, where the QC region gradually broadens with increasing temperature, here the transition to a quasiparticle metal occurs immediately after the $e_g$ gap closure and spin crossover onset, leading to a Brinkman-Rice line with weak temperature dependence and an abridged pressure extent for the QC "fan" at high temperatures. Orbital selectivity and the associated spin crossover phenomena thus dramatically affect the form of the insulator-metal transition behavior at these very high temperatures, producing markedly weak temperature dependence of all physical quantities within the QC region.

We relate our findings to existing knowledge on the experimental phase diagram of FeO by presenting our results as a function of pressure, where pressure is calculated at each volume-temperature condition using the experimentally determined equation of state for $B1$-FeO[46], as shown in Fig. 4. Here we include experimentally estimated phase boundaries for different crystal structures[46], as well as the melting curve[48]. We note that the phase coexistence region, where both insulating and metallic phases are present at $T < T_c$ ~370 K (omitted in Fig. 4, see Fig. 1), is predicted to lie at the center of the experimentally estimated stability field for rhombohedrally distorted $rB1$-FeO. In addition, we observe that the Brinkman-Rice line below ~2000 K,

marking the onset of an orbitally selective metal, traces the experimentally reported $B1$-$B8$ transition boundary. These observations raise further questions regarding the relationship between insulator-metal transitions and crystal structures in strongly correlated systems, which merit further investigation but are beyond the scope of this study.

## Consequences for transport properties

The three electronic phases identified for FeO in this study exhibit highly distinct transport properties (Fig. 4). Conductivity in the insulating state is relatively low (~$10^0$–$10^3$ S/m) and increases with temperature, as expected for thermal activation. In the correlated metallic state, conductivity is large (~$10^6$–$10^8$ S/m) and decreases with increasing temperature. In contrast, conductivity in the QC state lies at intermediate levels (~$10^4$–$10^5$ S/m) and displays remarkably weak dependence on both pressure and temperature. As discussed above, transport in the QC state is a consequence of a (poorly) conducting $t_{2g}$ band that lacks the presence of coherent quasiparticles. Unlike in a quasiparticle metal, where the mean-free path for electron-electron scattering is generally much longer than the lattice spacing, conductivity in the QC state lies around the Mott-Ioffe-Regel (MIR) limit (~$10^5$ S/m) characterized by a short mean-free path comparable to the lattice spacing[38]. Physically, the electrons exhibit Brownian-style diffusive motion caused by strong and frequent scattering.

## Robustness of theoretical results

The theoretical results we have obtained reveal that at temperatures on the scale of thousands of Kelvin, the insulator to metal crossover displays a significant intermediate regime, in close analogy to what is generally expected for quantum criticality[34,49,50]. Although there are several aspects of our work that may shift the precise location of the crossover, the general topology of the phase diagram would not be

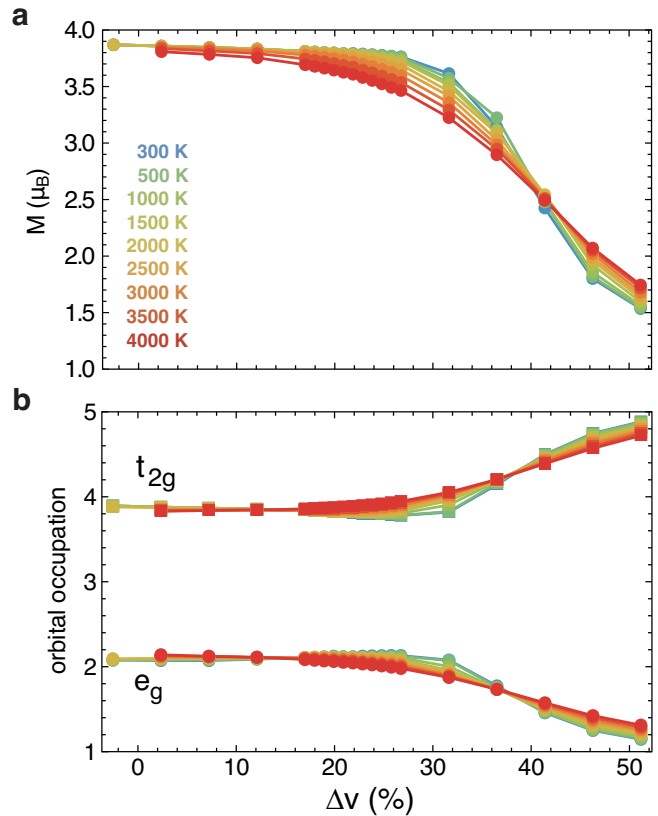

**Fig. 3 | Spin crossover behavior of *B*1-FeO at high *P–T* conditions.**
**a** Instantaneous local magnetic moment *M* results show that Fe remains in the high spin state throughout the insulator-metal transition region, with the onset of spin crossover only upon further compression with the closure of the $e_g$ gap. **b** The spin crossover reflects partial charge transfer from the $e_g$ to the $t_{2g}$ orbital, with remarkably weak *T*-dependence.

affected. Our result was obtained for specific values of the interaction parameters *U* and *J*, which we fixed to the values expected under ambient conditions[51]. We did so to avoid deliberate "fine-tuning" of input parameters, although we do expect that these interactions should display some volume/pressure dependence. Still, these details should not affect the qualitative and even the semi-quantitative aspects of our results. Similarly, the presence of small concentrations of Fe vacancies or small amounts of Mg substitutions could slightly displace the crossover line positions. We emphasize, however, that the characteristic scale of the electrical conductivity set by the Mott-Ioffe-Regel limit in the QC regime (-10⁵ S/m) should be a robust feature of our results. In particular, the modest pressure and temperature dependence of transport in the QC regime suggests that small shifts in the crossover line positions due to the effects discussed above will not affect the key finding that FeO exhibits intermediate values of electrical conductivity (-10⁵ S/m) at lowermost mantle conditions. Furthermore, various other physical mechanisms (such as different forms of magnetic order) that often play out at low to ambient temperatures ($T$ -10¹–10² K) are expected to be negligible at the $T$ -10³ K levels that we consider here. In this sense, the single-site DMFT theory we adopt, which deliberately ignores such magnetic correlations, should be regarded as an accurate solution to the electronic many-body problem under conditions relevant to Earth's interior.

## Comparison to previous results
We quantitatively compare general trends and magnitudes of transport obtained from our theoretical calculations to previous experimental measurements. A shock compression study also reported

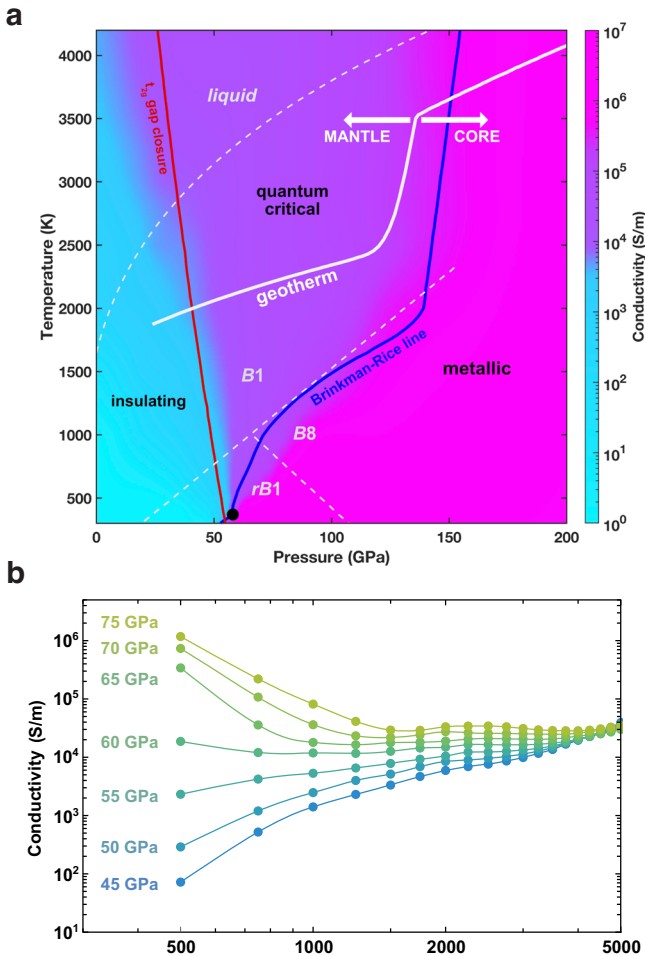

**Fig. 4 | Conductivities of *B*1-FeO. a** Theoretical phase diagram for *B*1-FeO, as a function of pressure calculated from volumes using the experimental thermal equation of state[46]. Color-coded are calculated values for electrical conductivity $\sigma$, which span only about one order of magnitude within the entire QC region at magnitudes comparable to the MIR limit (-10⁵ S/m) in other Mott oxides[38]. Solid and dashed white lines show the geotherm[84,85] and experimentally estimated phase boundaries[46,48], respectively. **b** Characteristic "fan-like" evolution of temperature-dependent conductivity curves is shown for 45 GPa ≤ *P* ≤ 75 GPa, as expected for Mott quantum criticality[34,36]. Note the markedly weak pressure and temperature dependence of the resistivity at *T* > 2000 K.

conductivities on the order of 10⁵·⁵–10⁶ S/m for pressures between 72 and 155 GPa and at elevated but unconstrained temperatures[52]. Static compression experiments reported weak temperature dependence of electrical resistance when heating up to -2500 K in the pressure range ~40–80 GPa and when heating above ~2000 K from 80 to 125 GPa[29]. The QC region determined in our study spans these *P–T* conditions and provides a physical basis for the observed weak temperature dependence. Our findings suggest that very shallow minima in resistivity-temperature measurements from these experiments should not be interpreted as marking the location of a sharp insulator-metal transition but could stem from secondary effects, such as phonon (lattice) interactions or defect mobility. In addition, the same experiments reported a plateau in conductivity at around 10⁵ S/m along a pressure range of ~60–120 GPa for *T* = 1850 K[29] (Fig. 5). These measurements of a conductivity plateau (weak pressure dependence) and magnitudes around the Mott-Ioffe-Regel limit (-10⁵ S/m) (Fig. 5) are now clearly explained and supported by the global phase diagram determined in this study, and in particular by the existence of the QC region. Overall, the electronic phase diagram and consequent

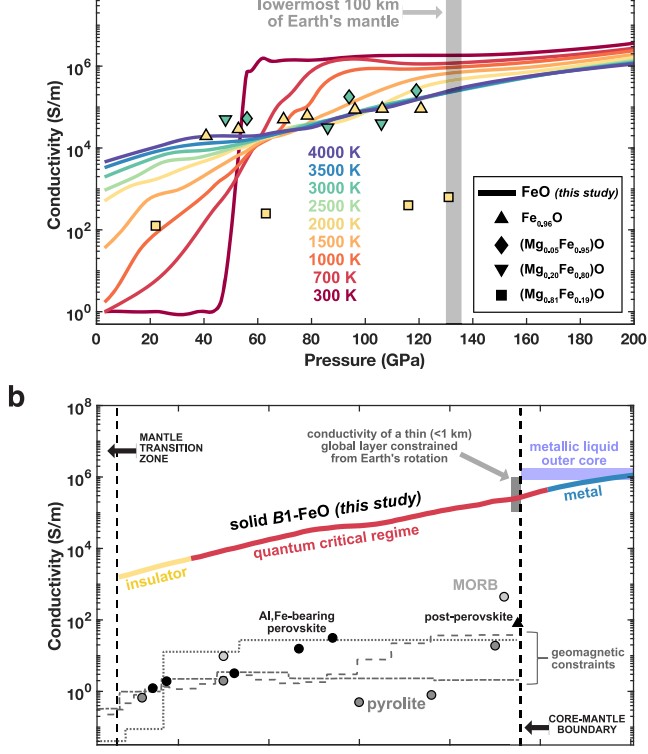

**Fig. 5 | Electrical conductivity of FeO compared with other Earth and planetary materials. a** Solid lines are calculated in this study, while symbols show previous electrical conductivity experiments on $Fe_{0.96}O$[29], $(Mg_{0.05}Fe_{0.95})O$ and $(Mg_{0.20}Fe_{0.80})O$[30], and $(Mg_{0.81}Fe_{0.19})O$[5], all color-coded by temperature.
**b** Conductivities as a function of depth in the Earth along the geotherm. Solid line is FeO (this study); gray lines show geomagnetic constraints on bulk mantle electrical conductivity (dashed[58], dotted[59], dot-dash[57]); points show experimental reports for bridgmanite (perovskite)[4], post-perovskite[54], pyrolite, and MORB[56]; blue shading shows theoretical reports for liquid iron alloys[7,60,61]; gray shading shows the conductivity range of a proposed thin layer at the mantle base[65].

transport properties determined for $B1$-FeO in this study provide a clear physical explanation for experimental reports on the material's conductivity. Our theoretical results capture much of the same features as those reported in previous theoretical works performed by using DFT+DMFT methods for $B1$-FeO[29,44,45,53], although our expansive canvassing of the entire phase diagram provides qualitatively updated insight and interpretation. Specifically, we demonstrate that a clearly defined intermediate regime arises between the insulator and the metal, with distinct spectral and transport signatures.

## Implications for Earth's interior
We find that the electrical conductivity of FeO at lower mantle conditions exhibits intermediate values ($10^4$–$10^5$ S/m) relative to the insulating mantle and metallic core. The lower mantle features conductivity magnitudes of $10^0$ to $10^2$ S/m based on experimental and geomagnetic constraints. Experiments on the major lower mantle phases bridgmanite[4], ferropericlase[5], and post-perovskite[54] have reported conductivities from $10^0$ to at most $10^3$ S/m, similar to experiments on the hydrous silicate phase D[55], as well as on pyrolite and mid-ocean ridge basalt (MORB) rocks[56] that represent the average lower mantle and subducted oceanic crust, respectively (Fig. 5). These values are in good agreement with depth profiles for conductivity, determined from geomagnetic observations[57–59]. For the metallic outer core, theoretical computations have reported conductivities for liquid

iron alloys around $10^6$ S/m[7,60,61], similar to experimental measurements on solid iron and iron alloys at high $P$–$T$ conditions[6,62–64]. The intermediate conductivity values for FeO at lowermost mantle conditions (~$10^5$ S/m) are robust even for small amounts of Mg substitution (up to 20%), based on experimental results[30], suggesting that iron-rich (Mg,Fe)O in the lowermost mantle would exhibit a unique signature of electrical conductivity relative to coexisting materials.

Interestingly, the base of Earth's mantle has been suggested to exhibit a unique signature of moderate electrical conductivity, higher than the bulk mantle but lower than outermost core fluid (Fig. 5), that affects electromagnetic coupling of the mantle and core and thus Earth's rotation and magnetic fields. Specifically, variations in the length of day over periods of several decades, as well as nutations of Earth's rotation axis on the diurnal timescale, are best explained by a mantle basal layer 1 km thick with conductivity $10^5$ S/m[65]. Further, low temporal variations of Earth's magnetic field in the Pacific region have recently been attributed to a non-uniform conducting layer at the mantle base with higher conductance levels in the Pacific, estimated at $6$–$9 \times 10^8$ S compared to $10^8$ S for a global average layer[66]. This elevated conductance could be approximately explained by 20–30 km thick structures with conductivity ~$10^5$ S/m covering around one-third of the mantle base on the Pacific, compatible with typical heights and detection locations of ultralow velocity zones[14].

Separately, independent seismic observations of the mantle base combined with geodynamic and mineralogical constraints have recently shown that ultralow velocity zones can be quantitatively explained by highly FeO-rich solid material[24–26]. Geodynamic work has further suggested that these mountain-scale structures may form from a thin layer[23,67] that could be difficult to detect seismically[68]. The bulk conductivity of such features would depend on the interconnection of moderately conductive FeO in the assemblage, which is poorly constrained. However, the remarkably low viscosity of the material ($10^{12}$ Pa·s) at lowermost mantle conditions[28] and its relatively high abundance in ultralow velocity zones suggested by recent work (~20–40%)[8,25,26] supports the possibility of interconnected networks of iron-rich (Mg,Fe)O and resulting bulk conductivity similar to $10^5$ S/m.

A solid FeO-rich mineralogy could thus provide a unifying explanation for constraints on Earth's mantle base from both seismic imaging as well as independent observations of temporal variations in Earth's rotation and magnetic field. FeO-rich structures could further imply heterogeneous thermal conductivity at the core-mantle boundary, instead of homogeneous heat flow out of the core assumed in some models of mantle dynamics[69,70]. Using the Wiedemann-Franz law and our calculated conductivity of ~$2 \times 10^5$ S/m, we estimate an electrical contribution to the thermal conductivity of ~17 W/m·K for FeO at the core-mantle boundary. This value is around two to four times larger than the reported thermal conductivity of the average pyrolitic lowermost mantle[71]. By a direct calculation of the electrical and the thermal conductivity, recent work on Hubbard models suggested[72] that the Lorentz number defining the Wiedemann-Franz law should be somewhat lower at high temperatures, as compared to the conventional value. Nevertheless, our general conclusions should remain qualitatively valid. Solid FeO-rich ultralow velocity zones may thus represent regions of high thermal conductivity at Earth's mantle base, which could promote the generation of long-lived mantle plumes, influence convection dynamics, and affect crystallization processes in the core[73–75].

## Methods
The eDMFT algorithm we use[40–42] starts with the calculation of the eigen-energies and the eigen-wavefunctions of the crystal by solving the DFT equations. Next, the correlated orbital subset is projected out as "quantum impurities" by a real-space projectors without downfolding, while the uncorrelated orbitals are treated by DFT, and act as a mean-field bath on the quantum impurities, resulting in a hybridization

between the two. The hybridization functions are determined self-consistently by solving the DMFT equation. The quantum impurities are solved by the hybridization expansion continuous time quantum Monte Carlo (CTQMC)[76,77] method. The modified charge density derived from combined DFT and DMFT equations is then used as the input of the next DFT iteration. The eDMFT algorithm iterates until full convergence of the charge density, the impurity self-energies, and the lattice Green's function etc are achieved. Finally, the maximum entropy method[78] is employed to analytically continue the Green's function and the self-energy from the Matsubara frequency to the real frequency axis. The linear augmented plane wave method is used as a basis, as implemented in WIEN2K package[79], and the local density approximation (LDA)[80] to the exchange and correlation functional is employed in the DFT part. We use a double-counting scheme[81] which is known to be exact within the LDA exchange and correlation functional. In each DMFT iteration a huge number ($\sim 2.8 \times 10^{10}$) of Monte Carlo updates is used to reduce the statistical error. A Monkhorst-Pack mesh of at least $12 \times 12 \times 12\, k - $ points is used in the calculation. At the ambient pressure the energy window for projection of the correlated states is $\pm 10$ eV around the Fermi energy. At high pressure the energy window is expanded so that the same number of bands are included for projection as done at ambient pressure. Only the Fe-$3d$ electrons are treated as correlated with Coulomb interaction $U = 10.0$ eV and Hund's coupling $J = 1.0$ eV, which is based on previous constrained DMFT calculations of FeO at ambient pressure[51]. Throughout the paper we fix the Coulomb interaction $U$ and the Hund's coupling $J$ as volume independent. Although increased pressure should reduce $U$ and $J$ in real FeO material, it will only quantitatively tune the results in the paper, such as the exact position of the insulator-metal transition.

## Data availability
The theoretical data presented in the figures can be found in the Source Data files, which are provided with this paper and in a Zenodo data repository (https://doi.org/10.5281/zenodo.10307816). The full set of theoretical data generated during this study are available from the corresponding author upon reasonable request. Source data are provided with this paper.

## Code availability
The eDMFT code we utilized in this paper was developed by Kristjan Haule, and it is publicly available from Kristjan Haule's website (http://hauleweb.rutgers.edu/tutorials/index.html). This site also has the appropriate tutorials associated with this code.

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

## Acknowledgements

Work in Florida (W.D.H. and V.D.) was supported by the NSF Grant No. DMR-1822258, and the National High Magnetic Field Laboratory through the NSF Cooperative Agreement No. DMR-1644779 and the State of Florida. Work at Rutgers (K.H.) was supported by the NSF grant No. DMR-2233892. P.Z. acknowledges the support of NSFC grant No.11604255. J.M.J. and V.V.D. are grateful to the National Science Foundation's Collaborative Study of Earth's Deep Interior (EAR-2009935) and Geophysics (EAR-1727020) programs for support of this work. A portion of this work benefitted from the MINUTI open-source software, https://www.nrixs.com.

## Author contributions

V.D. designed the project. V.V.D. and J.M.J. provided the geophysical context and implications. W.D.H. and P.Z. equally contributed to the computational work and data analysis. K.H. provided the eDMFT code and technical guidance. W.D.H. and V.V.D. produced the figures. V.V.D. and V.D. wrote the original manuscript. All authors discussed the results and commented on the manuscript.

## Competing interests

The authors declare no competing interests.
