## [Peer Review File · Nature Communications]

Quantum critical phase of FeO spans conditions of Earth's lower mantleEditorial Note: Figure 1 on page 13 of the Peer Review File has been redacted as indicated to remove third-party material where no permission to publish could be obtained.

REVIEWER COMMENTS

Reviewer #1 (Remarks to the Author):

The paper investigates FeO at conditions relevant to Earth's lower mantle using dynamical mean-field theory in a realistic setting. It evaluates the transport properties of the compound for a wide range of temperatures and pressures, thereby depicting a global electronic phase diagram that documents the generic coherence-incoherence crossover as well as some specific aspects related to orbital selectivity in this compound.

This interesting paper tells that FeO is in the high-temperature quantum critical regime. This finding, and the discussion of the relation of it to the geophysical picture of Earth's interior are in my opinion interesting to a broad community that includes material scientists, geophysicists, and workers in the field of strong electronic correlations.

The methodology is robust and the paper overall supports its conclusions well. In my opinion the paper should be published and Nature Communications is appropriate journal for its content. Below I address few aspects where I believe the discussion can be improved.

1. The authors say that conductivity in quantum critical state is about MIR limit. But, in principle, at high temperatures there is no bound to conductivity, as seen also in their figure 4 at high temperatures. Is the statement then that the compound coincidentally is at conditions corresponding to that value of resistivity or is there something more profound? Does the QC regime end at temperatures sufficiently high?

2. In the geophysical discussion part, the paper tells that earlier work supports existence of a thin layer of Fe oxides with intermediate values of conductivities 10^5 S/m. I did not understand why authors say "This finding is particularly intriguing...". Is not the content of both paragraphs, that is the one above and that sentence, consistently understood in terms of iron oxides with that values of conductivity?

3. What precisely is the geophysical value of the presented results? Are not the values of conductivities known from earlier work (e.g. Ohta et al'12)? [I understand that earlier this was not discussed in terms of quantum criticality which is I think an indisputable new contribution here, and realizing it is important for geophysics from qualitative aspects but does it really matter for geophysical modelling also from the practical point of view ?]

4. Authors use the Wiedemann-Franz law for a case of electron-electron dominated resistivity in a high-T regime where it is likely to fail, see e.g. PRB 106 245123 Fig 6 for a recent example. Furthermore in Ohta et al., the experimental values of conductivities seem lower. Both effects alone can easily reduce the inferred thermal conductivity by a factor of 2. Given that calculations of thermal conductivity are easily accessible to authors directly, what do they get?

5. Authors used some choice for interaction parameters, which is reasonable in absence of exact knowledge of them. However, I am not convinced by the formulation: 'We did so to avoid deliberate "fine-tuning"'. Do the authors want to say that such a fine-tuning would be meaningful (e.g., the experimental input is sufficient?)

Perhaps instead of saying this it would be more valuable for the reader if the authors explained why do they prefer larger values of interaction parameters than Leonov et al? To what extent are the resistivities affected if these smaller values $U=7, J=0.9$ are used?

6. Could word "strong" be removed here and there?

"strongly match" -> "match"

"strong explanation" -> "explanation"

Reviewer #2 (Remarks to the Author):

Ho et al. study the electronic structure of FeO at the P-T region relevant to the core-mantle boundary. They reported a quantum critical phase between Mott insulator and correlated metal and computed the conductivity for this phase.

While the author argued for the significant geophysical implications that have been covered in previous studies, the major results of the work are not new.

- For instance, the "quantum criticality" has been identified for a series of 3d metal oxides, including FeO, in Ref. 50. The author just expands the volume and temperature with much more calculations.

- The spin crossover behavior has also been computed by some of the coauthors with the same method in [Journal of Physics: Conf. Series 827 (2017) 012006, doi:10.1088/1742-6596/827/1/012006]. The results also seem qualitatively the same. (Why is this paper not cited?)

- The only geophysical relevant data provided by the present study - the conductivity, has also been computed in Ref. 9 by one of the coauthors and the same method. The results are almost the same $\sim 10^5$ S/m near CMB.

Therefore, this work did not provide much new understanding of FeO-rich LLSVZ.

The authors did not quantitatively compare their results with other calculations. It's very hard to judge their validity.

- For example, there is a recent work on FeO that shows a strong temperature dependence HS-LS transition [Greenberg et al. arXiv:2004.00652]. It was computed with the same DFT+DMFT, but the present work shows a weak temperature dependence. The B1 equation of state was computed by Greenberg et al.. The authors should also make a comparison to it.

- Previous calculations showed that the U parameter is very sensitive to the pressure for FeO [Sun et al. Phys. Rev. Materials 4, 063605]. But the authors used a fixed U parameter and claim it does not change the main results.

Overall, this paper does not provide many novel results to justify a high-impact publication.

Reviewer #3 (Remarks to the Author):

The authors present a comprehensive computational study of electronic properties of FeO using a state of art method for description of correlated materials. The article aims at an interdisciplinary audience as its goal is an improved understanding of geophysical implications of FeO physics. I am able to comment on physical aspects of the present work.

I do not have any major criticism, but rather several remarks concerning the presentation of the results.

1) The use of 'quantum critical'. The authors denote part their phase diagram as a QC (quantum critical) region and use the term also in title. In the text they introduce quantum critical in quotes. The term 'quantum critical' has an established meaning in physics referring to quantum fluctuations dominating over thermal fluctuations and usually connected to a phase transition at zero temperature. I am not sure that it is case here. The authors should better justify the use of QC in the text or resort to an different terminology.

2) In Fig. 3 the 'magnetic moment' M is plotted as a function of temperature. This is a paramagnetic phase therefore the definition of M should be given. Is it a Curie-Weiss moment determined from T-dependence of local susceptibility?

3) Fig. 5 presents the calculated conductivities as functions of pressure. The calculations were performed at fixed density. The authors should explain and present, e.g., in SM, how the volume \rightarrow pressure conversion was done, e.g., present energy vs volume curves.

4) The comparison to previous theoretical work is essentially absent. As far as I can say Ref. 50 does not show the orbital-selective QC regime. If my observation is correct, this represents a substantial difference to the previous DFT+DMFT results and should be pointed out explicitly. What is the possible origin of this discrepancy?

Is the orbital-selective QC 'phase' a realization of the state observed in 2-orbital model of PRL 99, 236404 (2007)?

5) In the methods description the author that 'The correlations treated by both the DFT and eDMFT are subtracted exactly [82]'. I find the word exactly misleading. Even in Ref. 82 (written by one of the present authors) the author used 'exactly' in quotes and the exactness is not generally accepted. I find some like 'double counting correction of Ref. 82' a more accurate description.

6) Work lacks many useful details, which should be provided at least in SM. i) Form of the interaction. Ref. 50 used density-density approximation. Slater-Kanamori or full-Coulomb interaction assuming spherical harmonic orbitals are other common choices, however, DMFT studies with less symmetric orbitals were also published. Therefore specific form should be given.

ii) Real-frequency self-energies for the presented spectra would be very helpful. iii) Energy vs volume curves should be accessible with the present data and are needed for volume \rightarrow pressure transformation. iv) Formulas. Besides the e-e interaction, the authors should provide definition of M and occupation numbers (describe projections on correlated orbitals), as well as the formula to compute dc conductivity.

A. Reviewer #1 (Remarks to the Author):

Reviewer's Comments: *The paper investigates FeO at conditions relevant to Earth's lower mantle using dynamical mean-field theory in a realistic setting. It evaluates the transport properties of the compound for a wide range of temperatures and pressures, thereby depicting a global electronic phase diagram that documents the generic coherence-incoherence crossover as well as some specific aspects related to orbital selectivity in this compound.*

Our reply: We thank the Reviewer for emphasizing the novelty of our work, namely, noting that we have calculated the global phase diagram of FeO for a wide range of temperature and pressures in a realistic setting, allowing us to depict the global phase diagram of the compound, something that has not been achieved before.

Reviewer's Comments: *This interesting paper tells that FeO is in the high-temperature quantum critical regime. This finding, and the discussion of the relation of it to the geophysical picture of Earth's interior are in my opinion interesting to a broad community that includes material scientists, geophysicists, and workers in the field of strong electronic correlations. The methodology is robust and the paper overall supports its conclusions well. In my opinion the paper should be published and Nature Communications is appropriate journal for its content. Bellow I address few aspects where I believe the discussion can be improved.*

Our reply: We thank the Reviewer for stressing that our methodology is robust, which convincingly addresses the objection of the second reviewer, who suggested that our results are sensitive to value of U chosen, which is not the case (see discussions below).

Reviewer's Comments: *1. The authors say that conductivity in quantum critical state is about MIR limit. But, in principle, at high temperatures there is no bound to conductivity, as seen also in their figure 4 at high temperatures. Is the statement then that the compound coincidentally is at conditions corresponding to that value of resistivity or is there something more profound? Does the QC regime end at temperatures sufficiently high?*

Our reply: The MIR limit is generally regarded not as a precise **bound** on the conductivity, but simply an order-of-magnitude estimate, corresponding to the mean-free path on the scale of the inter-atomic spacing. It is characteristic of the incoherent regime characteristic of the insulator-metal transition region, which in our case corresponds to the Mott quantum critical region. We agree with the reviewer that there exists no strict upper bound on the conductivity at high temperatures. Still, as we can clearly see from Fig. 4 (bottom panel), different resistivity curves essentially "merge" at high temperatures, towards a value corresponding to the MIR limit of around 10^5 S/m. As a result, transport displays extremely weak dependence on both pressure and temperature within the entire quantum critical regime, surrounding the geotherm trajectory. This fact is also very clearly seen in the color-coded phase diagram (Fig. 4, top panel).

We stress that very similar behavior is seen in many other examples of Mott quantum criticality such as Mott organics or the transition metal dichalcogenide (TMD) moiré materials, although in those cases the characteristic energy scales are by orders of magnitude smaller (the energy scale of the bandwidth in Mott organics is about 10^3 K, and in TMD moiré materials around 10^2 K, whereas the bandwidth of Mott oxides such as FeO is on the scale of 10^4 K).

Concerning the high-T end of any QC region, one generally expects it to extend to the scale comparable to the basic energy scales in the problem, in our case the bandwidth $W \sim 10^4$ K. Crystalline FeO we consider, however, cannot survive up to such high temperatures, because such oxides typically melt around 3,000 K - 4,000 K, as indicated in Fig. 4 (top panel). Nevertheless, the physically relevant P-T range surrounding the geotherm trajectory throughout the lower mantle remains within the crystalline (rocksalt B1) phase, and is entirely contained within the QC regime we have discovered.

Reviewer's Comments: *1.2. In the geophysical discussion part, the paper tells that earlier work supports existence of a thin layer of Fe oxides with intermediate values of conductivities 10^5 S/m. I did not understand why authors say "This finding is particularly intriguing...". Is not the content of both paragraphs, that is the one above and that sentence, consistently understood in terms of iron oxides with that values of conductivity?*

Our reply: We apologize if the original wording of this section was confusing. We have re-organized this part of the discussion for further clarity. As we clarify in the revised text, the main point here is that seismic observations of ultralow velocity zones (ULVZs) have led to numerous possible mineralogies proposed to explain the observed low velocities, each of which holds different implications for Earth's evolution. As these seismic observations of the mantle base have rapidly developed over the last decade, however, almost no previous work has connected those

developments to the completely independent set of observations constraining a moderately conductive thin layer at the mantle base. These electrical constraints provide an opportunity to test the viability of different mineralogies proposed to explain seismic observations. One key conclusion of this study is that recent cutting-edge results from mineral physics show that FeO-rich mineralogies can explain independent observations of both low seismic velocities and intermediate conductivities. This is now the first proposed mineralogy shown to quantitatively reproduce both sets of observations, and this conclusion is possible due to the new global phase diagram from this study.

Reviewer’s Comments: *3. What precisely is the geophysical value of the presented results? Are not the values of conductivities known from earlier work (e.g. Ohta et al’12)? [I understand that earlier this was not discussed in terms of quantum criticality which is I think an indisputable new contribution here, and realizing it is important for geophysics from qualitative aspects but does it really matter for geophysical modelling also from the practical point of view ?]*

Our reply: As Reviewer 1 states, values of electrical conductivity for FeO were indeed experimentally reported by Ohta et al. (2012). However, several limitations exist for these measurements. Firstly, these represent extremely challenging experiments with relatively high uncertainties due to difficulties in precisely quantifying sample geometries and temperature distributions within the samples. This raises questions around the accuracy and precision of the reported conductivity values, especially in the absence of any reports from other independent studies or insight into reproducibility of the results. This is particularly true given the weak P-T dependence of conductivities and conductivity magnitudes lower than those expected for a metal. These phenomena were not well understood in that original study, which attributed the results to a sharp insulator-metal transition. However, as the conductivity magnitudes were lower than those expected for a metal, this called into question the accuracy of the measurements and possible consequences for the Earth.

Additionally, the measurements in Ohta et al. (2012), while an important experimental achievement, span a relatively limited range of pressures and temperatures that prevent a full exploration of the global electronic phase diagram, as is done in this study. In particular, the experiments and theoretical calculations in that study did not reach conditions of Earth’s mantle base. Further, most of the reported experimental data was presented as raw measurements of electrical resistance, while actual values of electrical conductivity (calculated from uncertain sample geometry) were reported only at relatively low temperature (1850 K compared to ~ 3500 -4000 K expected for the mantle base) and for only seven pressure points. This leads to a lack of clear understanding how temperature and pressure influence conductivity in different regions of the phase diagram and thus presents a major obstacle to extrapolating reported values to P-T conditions of the mantle base. This is especially problematic given the weak P-T dependence of conductivity reported in that study but not understood well due to the absence of the Quantum Criticality framework. This is additionally limiting due to the fact that the low temperature considered in that study (1850 K) is below the temperature ~ 2000 K where the pressure-dependence of the Brinkman-Rice line changes dramatically due to the closure of the e_g gap. This provides yet another obstacle to confidently extrapolating previously reported values to conditions relevant for Earth’s mantle base.

The results in our study are thus critical for geophysical modeling for several reasons. This work provides a novel physical framework for understanding the electronic phase diagram of FeO at extreme conditions. In doing so, it supports the conductivity magnitudes reported by Ohta et al. (2012) and gives a new explanation for the observed P-T dependence of conductivity, distinct from either an insulator or a metal and relevant to Earth’s mantle base. Further, the global phase diagram determined in this study includes calculations of transport at the exact P-T conditions of Earth’s mantle base and quantifies effects of pressure and temperature variations at those conditions. This is a significant development beyond what was reported by Ohta et al. (2012).

Finally, as discussed in response to point 2 above, the last decade has seen great attention on rapidly growing seismic constraints for heterogeneities at the mantle base. Research from the past several years has provided new quantitative evidence for solid FeO enrichment as viable hypothesis for ultralow velocity zones. However, FeO enrichment is one of multiple proposed hypotheses, and it has been challenging to uniquely constrain mineralogies in the region based exclusively on seismic observations. The findings in this study on the conductivity of FeO are significant because they allow for the FeO-enrichment hypothesis to be additionally tested against fully independent constraints from geophysical observations of Earth’s rotation and magnetic field features. Our findings of unique intermediate conductivity values for FeO in the quantum critical region and at mantle base conditions show that the FeO enrichment hypothesis can explain independent observations of low velocities and elevated conductivities for the mantle base. This provides a new paradigm to understand the thermochemical state of the present-day mantle base and opens various implications for Earth’s evolutionary history.

Reviewer's Comments: 4. *Authors use the Wiedemann-Franz law for a case of electron-electron dominated resistivity in a high-T regime where it is likely to fail, see e.g. PRB 106 245123 Fig 6 for a recent example. Furthermore in Ohta et al., the experimental values of conductivities seem lower. Both effects alone can easily reduce the inferred thermal conductivity by a factor of 2. Given that calculations of thermal conductivity are easily accessible to authors directly, what do they get?*

Our reply: We thank the Reviewer for this important point, and for pointing out the significant works of Mravlje et al. (e.g. PRB 106, 245123(2022)), which clearly demonstrate the violation of the strict Wiedemann-Franz law at high temperatures. In our paper we utilized the Wiedemann-Franz law to simply obtain a very rough (order of magnitude) estimate of the thermal conductivity, based on our results for the electrical conductivity. We agree with the Reviewer that, as a matter of principle, our methodology can be utilized to also compute the values for the thermal conductivity, without using the Wiedemann-Franz law. Performing such demanding calculations across the phase diagram is, however, beyond the scope of this paper, and is left for future work. Based on the useful results of PRB 106, 245123(2022), we infer that such more accurate computations should produce slightly smaller values of the thermal conductivity at the core-mantle boundary, compared to the estimates we obtained using the Wiedemann-Franz law. However, our conclusion should remain qualitatively valid. We make this point in the revised manuscript and have added the corresponding reference to the reference list.

Reviewer's Comments: 5. *Authors used some choice for interaction parameters, which is reasonable in absence of exact knowledge of them. However, I am not convinced by the formulation: 'We did so to avoid deliberate "fine-tuning"'. Do the authors want to say that such a fine-tuning would be meaningful (e.g., the experimental input is sufficient?)*

Our reply: Fine tuning could, of course, make the agreement with experiment better, but only for by a few percent, if we use state-of-the-art computational estimates of U within the current method. This method is a self-consistent "constrained-DMFT" calculation, which requires one to construct a rather large unit cell of the same compound using the same eDMFT techniques we utilize. In this approach, one atom would need to have constrained occupations, while all other sites in the solid are allowed to screen the charging energy on this chosen site (the "impurity"). We previously calculated this value at ambient pressure, which gives us our choice of U , but we did not repeat this demanding calculation as a function of pressure, since its change was estimated to be weak and insignificant for the present study. The more accurate task of microscopically estimating $U(P)$ is left for further work.

From a general perspective, however, there remains a generic issue of accurately determining the screening in solids, which is a rather difficult problem, because it depends strongly on the type of orbital used in DMFT calculation, and also on including/neglecting the vertex corrections. Namely, most of the DMFT groups tend to use DMFT orbitals which are much more extended in real space, but much less extended in energy range. These orbitals are typically produced by "Wannierization" of a few selected bands, hence they are strongly constrained in energy, and therefore not that well-localized. In such alternative DMFT methods, the total energy is much harder to compute to high precision, hence these groups typically do not employ the accurate self-consistent "constrained-DMFT" calculation we utilize to determine U , but rather rely on so-called "constrained RPA". The latter is a one-loop calculation (response of the non-interacting system), which neglects vertex corrections. One can show that this is equivalent to calculate fluctuations (linear response) within the Hartee-Fock approximation, i.e. this method is very different from our constrained DMFT, which is a many-body method and includes DMFT vertex corrections.

Reviewer's Comments: *Perhaps instead of saying this it would be more valuable for the reader if the authors explained why do they prefer larger values of interaction parameters than Leonov et al? To what extent are the resistivities are affected if these smaller values $U=7, J=0.9$ are used?*

Our reply: The smaller values of U and J used by Leonov et al. are related to their very different choice of localized orbitals in DFT+DMFT calculations. They generally use more extended orbitals, and hence the values of U, J are not transferable from one method to another. The results would definitely be worse when such incompatible values of U and J are used.

Reviewer's Comments: 6. *Could word "strong" be removed here and there? "strongly match" → "match" "strong explanation" → "explanation"*

Our reply: We thank the Reviewer for this suggestion. We have edited the manuscript to reduce repetition of the word "strong", particularly in discussion of the results and implications.

B. Reviewer #2 (Remarks to the Author):

Reviewer’s Comments: 1. *Ho et al. study the electronic structure of FeO at the P-T region relevant to the core-mantle boundary. They reported a quantum critical phase between Mott insulator and correlated metal and computed the conductivity for this phase.*

While the author argued for the significant geophysical implications that have been covered in previous studies, the major results of the work are not new.

Our reply: We respectfully but strongly disagree. No previous work has precisely defined or explicitly characterized the quantum critical region associated with the Mott point in FeO. Our viewpoint is reinforced by the comments of Reviewer 1 that our work is “...interesting to a broad community...” and our “...methodology is robust and the paper overall supports its conclusions well...”. Reviewer 3 also thinks that we have done “...a comprehensive computational study of electronic properties of FeO using a state of art method...”

Reviewer’s Comments: 2. - *For instance, the “quantum criticality” has been identified for a serial of 3d metal oxides, including FeO, in Ref. 50. The author just expands the volume and temperature with much more calculations.*

Our reply: We agree with the Reviewer that some of the recent papers of Leonov et al. indeed mention the concept of “quantum criticality”, but they never precisely define what is meant by this term, nor what is its physical content. In fact, while PRB 101, 245144 (2020) talks about quantum criticality in its title, it barely talks about it within the paper itself. The paper neither defines precisely what is the boundary of the QC region, nor does it explain its physical distinction from the other regimes in the phase diagram. In addition, the same author published several earlier papers on Mott oxides, including FeO, with predictions/results that are not only at odds with our quantum critical scenario, but even with his own more recent results. Here they claimed that much of the relevant regime should display a first order transition and phase coexistence due to volume collapse up to elevated temperatures of thousands of Kelvin. We carefully checked for this possibility within our state-of-the-art scheme, but find no evidence for volume collapse at elevated temperature. We suspect that the somewhat contradictory messages offered by the series of Leonov et al. papers simply reflect the limitations of the accuracy of their theoretical scheme and the computational refinements they were able to achieve, leading to inconclusive results concerning these subtle issues.

The series of papers by Leonov et al., although recently recognizing the importance of the concept of quantum criticality that we advocate, have not provided any convincing description of the phenomenon. In contrast, in our work we not only define and quantitatively calculate the boundaries of the QC regime, but also clearly explain its essential difference from the other regimes. Comparing with the Mott insulator regime with a spectral gap and deactivated transport, and the coherent quasiparticle regime on the metallic side, the QC regime has neither the well-defined gap, nor the coherent quasiparticle states leading to high mobility. We have also stressed the similarities and differences to other Mott systems known to display features of quantum criticality, thus revealing the deep connections between FeO and similar Mott oxides and other physical systems.

In short, Leonov et al. did not address the important issues relating to quantum criticality nor its significance for the geophysics of the deep Earth. In contrast, the message we put forward is novel, precise, and detailed, both in comprehensively defining the quantum critical regime and in clearly explaining its major implications for geoscience.

Reviewer’s Comments: 3. - *The spin crossover behavior has also been computed by some of the coauthors with the same method in [Journal of Physics: Conf. Series 827 (2017) 012006, doi:10.1088/1742-6596/827/1/012006]. The results also seem qualitatively the same. (Why is this paper not cited?)*

Our reply: In the 2017 paper [Journal of Physics: Conf. Series 827 (2017) 012006] the calculations were limited to only two temperatures 300 and 1000 K. The central idea presented in current manuscripts, FeO in Earth’s deep mantle is in the quantum critical states, is totally absent in it. To reveal the full relevance/role of spin crossover around the Mott insulator-metal transition, especially in the context of quantum criticality, we had to perform large-scale canvassing of the phase diagram with a fine mesh of data points in our current research. Nevertheless, we thank Reviewer for noting this previous paper, which we included as Ref. 52 in the revised manuscript.

Reviewer’s Comments: 4. - *The only geophysical relevant data provided by the present study - the conductivity, has also been computed in Ref. 9 by one of the coauthors and the same method. The results are almost the same 10^5 S/m near CMB. Therefore, this work did not provide much new understanding of FeO-rich LLSVZ.*

Our reply: We respectfully disagree, and refer Reviewer 2 to our responses to Reviewer 1 regarding the novelty of this work in its contribution to advancing understanding of Earth’s interior. As we discuss in response to point

3 from Reviewer 1, the previous study did indeed report conductivity values from experiments and theoretical calculations that are confirmed by results from this current study. However, the previously reported measurements and calculations were done across a limited range of pressures and temperatures. In fact, electrical conductivities were reported only at one temperature (1850 K) that is much lower than temperature of Earth's core-mantle boundary (3500 – 4000 K). This low temperature further falls below the temperature of ~ 2000 K where the pressure-dependence of the Brinkman-Rice line changes drastically due to the closure of the e_g gap. This presents a major obstacle to extrapolating conductivities determined at low temperatures to the high-temperature conditions of the lowermost mantle. Additionally, the mild temperature and pressure dependence of conductivity and the magnitudes of conductivity lower than those of a metal were not clearly understood in the previous study due to the lack of the quantum criticality framework. This led to questions about the accuracy of the electrical measurements and major difficulty in extrapolating those results to P-T conditions of the lowermost mantle.

Our study provides several key advancements that are critical for understanding the thermochemical state of Earth's core-mantle boundary. In particular, the global phase diagram calculated in this study across a wide pressure and temperature range provide a new unified understanding of electronic processes across deep Earth conditions. This framework confirms the previously reported values and provides a novel explanation for the reported P-T dependence, as well as extends calculations to pressure and temperature conditions relevant to Earth's mantle base and quantifies the effects of pressure and temperature variations in that region on conductivities.

Finally, as discussed in response to Reviewer 1, research over the past decade has provided new quantitative evidence that enrichment of solid FeO at the mantle base can explain seismic observations of the sound velocities and morphologies of ULVZs. However, determining a mineralogy that can uniquely explain observations is challenging, and several mineralogies have been proposed, each with different implications for Earth's present-day state and evolution. The findings in this study on the conductivity of FeO are significant because they allow for the FeO-enrichment hypothesis to be additionally tested against fully independent constraints from geophysical observations of Earth's rotation and magnetic field features. Our findings of unique intermediate conductivity values for FeO in the quantum critical region and at mantle base conditions, a result of the new global phase diagram, show the FeO enrichment hypothesis can explain independent observations of low velocities and elevated conductivities for the mantle base. This provides a new paradigm to understand the thermochemical state of the present-day mantle base and opens various implications for Earth's evolutionary history.

Reviewer's Comments: 5. *The authors did not quantitatively compare their results with other calculations. It's very hard to judge their validity.*

Our reply: We did compare our results with Ref. 9 and, as it was pointed out, we obtained the same results, as should be expected. However, different groups (e.g. Leonov et al.) use different DFT+DMFT methodologies, and most of them are known to be less reliable, as they use more extended Wannier orbitals. The negligence of non-local interactions in such more extended Wannier orbitals is a less reliable approximation.

Reviewer's Comments: 6. *- For example, there is a recent work on FeO that shows a strong temperature dependence HS-LS transition [Greenberg et al. arXiv:2004.00652]. It was computed with the same DFT+DMFT, but the present work shows a weak temperature dependence. The B1 equation of state was computed by Greenberg et al.. The authors should also make a comparison to it.*

Our reply: Again, quantitative comparison is difficult because Greenberg et al. do not utilize the same DFT+DMFT method we utilize, which is generally true for all the previous papers from the Leonov group. In contrast to our "embedded DMFT" (eDMFT) scheme, most others utilize the "downfolding" scheme based on Wannier functions. This older strategy is currently believed to be not as accurate as the modern eDMFT scheme, especially in applications over broad pressure/temperatures regimes of extreme conditions that we focus on in this study.

In contrasting the two methods, we can be even more precise. Although the work by Greenberg et al. uses a fully charge self-consistency as we did, there are important difference in the implementation details between the two methods. First, they choose a plane-wave pseudopotential approach for the solution of the Kohn-Sham equation, while we use the all-electron linearized augmented plane-wave as implemented in Wien2k. The latter offers very precise charge density near the nucleus and consequently very precise quasi-atomic orbitals. Second, in Greenberg et al. the correlated Fe-3d orbitals are represented in the Wannier function basis using narrower energy window around the Fermi level, while we are working in a more localized LAPW basis with much broader energy window, and within an all-electron calculation. As a result, Greenberg et al. must be using very different projectors from ours, to connect the local correlated orbitals with the bands of the Kohn-Sham eigenfunctions. We stress that in pseudo-potential methods the exact charge-density near the nucleus is not available, and high-quality local projectors are not easy to

construct. Unfortunately, the details of projector are not presented in arXiv:2004.00652, so we cannot make further comparison. In short, the physical differences between the two methods may be expected to lead to very different temperature dependence for the HS-LS transition. Since we are using the more precise all-electrons LAPW in DFT, as well as the improved LAPW and the high quality projector, our results should be more accurate.

Reviewer’s Comments: 7. - *Previous calculations showed that the U parameter is very sensitive to the pressure for FeO [Sun et al. Phys. Rev. Materials 4, 063605]. But the authors used a fixed U parameter and claim it does not change the main results.*

Our reply: Again, we respectfully disagree. When the local Hamiltonian (defining U) is based on low energy Wannier orbitals (spanning a small energy range) it is true that the results are very sensitive to the precise value of U . But this is definitely not true within our formulation for very localized quasi-atomic orbitals.

One way to understand this dichotomy is to think of the relation of the Hubbard model vs. the p-d model for cuprates. The Hubbard model would be the one with Wannier orbitals more constrained in energy space and more delocalized in real space, and the p-d model is the one where the d-orbital is allowed to be more localized in real space, but more extended in energy space. The solution of the Hubbard model is, of course, extremely sensitive to the value of U , as the position of the Hubbard bands is proportional to U . However, in the p-d model, the value of U is much larger, and the results of the p-d model are only weakly dependent on U , as the position of the Hubbard band is determined by charge-transfer processes, and hence hybridization with the oxygen and position of the oxygen levels (other orbitals) is the most important parameter.

Similar physics is going on here. Our calculation is more similar to generalized p-d model, while those by most other groups are closer to the generalized Hubbard model. Hence the difference in value of the parameter U we respectively use, and some difference in the physical observable. We do want to point out that, from a general perspective, the DMFT approach all groups utilize is a much better approximation in p-d models, as the Coulomb repulsion on the d orbital of the p-d model is much more localized (non-local U much smaller) than in the Hubbard model. This is why eDMFT implementation is generally far superior than earlier versions based on Wannier function approaches.

Reviewer’s Comments: 8. *Overall, this paper does not provide many novel results to justify a high-impact publication.*

Our reply: Again, we respectfully disagree. As explained above, our work goes far beyond any of the related efforts from previous works. We not only utilized a technically superior theoretical method and carried out a much more detailed canvassing of the phase diagram, but also were able to interpret these results within a novel physical picture of direct relevance to experiments.

C. Reviewer #3 (Remarks to the Author):

Reviewer’s Comments: *The authors present a comprehensive computational study of electronic properties of FeO using a state of art method for description of correlated materials. The article aims at an interdisciplinary audience as its goal is an improved understanding of geophysical implications of FeO physics. I am able to comment on physical aspects of the present work.*

I do not have any major criticism, but rather several remarks concerning the presentation of the results.

Our reply: We thank the Reviewer for recognizing our theoretical approach utilizes state-of-the-art methodology in performing a comprehensive study of this important mineral. This is an important point since, as we also pointed out in our reply to Reviewer 2, other recent works on FeO utilized alternative theoretical implementations of the DFT+DMFT approach, which are currently regarded as significantly less accurate for first-principle modeling of correlated matter.

We also thank Reviewer 3 for making it clear that he/she "...does not have any major criticism, but rather several remarks concerning the presentation..." Clearly, this statement speaks loudly of the validity of both our methodology and our major conclusions. The Reviewer simply wants us to slightly improve our presentation, which we have followed in great detail in the revised version of the manuscript. We are thankful to the Reviewer for these very

useful comments, which will make the paper much more accessible to the broad readership of Nature Communications.

Reviewer's Comments: 1) *The use of 'quantum critical'. The authors denote part their phase diagram as a QC (quantum critical) region and use the term also in title. In the text they introduce quantum critical in quotes. The term 'quantum critical' has an established meaning in physics referring to quantum fluctuations dominating over thermal fluctuations and usually connected to a phase transition at zero temperature. I am not sure that it is case here. The authors should better justify the use of QC in the text or resort to an different terminology.*

Our reply: We agree with the Reviewer that the concept of Quantum Criticality has an established meaning in physics, which is typically associated with a given zero temperature (quantum) phase transition. The concept was originally introduced in describing the onset of various magnetic and/or other types of orders in correlated metals, but more recent works on various systems has demonstrated that it equally well applies to the vicinity of the Mott point. Indeed, in systems such as Mott organic materials, as well as certain moiré materials, behavior analogous to what we advocate for FeO has been established in recent works, both theoretically and experimentally. In all these systems, as well as in FeO, the Mott point is a sharp quantum phase transition at $T=0$, which broadens into a wedge-like region at elevated temperature. This behavior, which is indeed dominated by quantum fluctuations of charge at the brink of delocalization, is precisely what one generally expects for quantum criticality. This is why we believe that describing the novel intermediate regime around the Mott point as a quantum critical region is perfectly justified and conceptually important. In fact, making this connection in the framework of geophysics is one of the key ideas of our paper. We thank the Reviewer for stressing the need to better explain this idea to the reader, as we have done in several places in the revised manuscript.

Reviewer's Comments: 2) *In Fig. 3 the 'magnetic moment' M is plotted as a function of temperature. This is a paramagnetic phase therefore the definition of M should be given. Is it a Curie-Weiss moment determined from T -dependence of local susceptibility?*

Our reply: It is the fluctuating magnetic moment, which is very closely related to the Curie-Weiss moment. We have added precise definition to supplementary information. We thank the Reviewer for asking us to better explain the physical meaning of this important quantity, as we now do in detail, by adding a new section to the Supplementary Materials.

Reviewer's Comments: 3) *Fig. 5 presents the calculated conductivities as functions of pressure. The calculations were performed at fixed density. The authors should explain and present, e.g., in SM, how the volume \rightarrow pressure conversion was done, e.g., present energy vs volume curves.*

Our reply: Pressures are calculated from the volume (density) and temperature conditions of each calculation using a (experimentally-determined) thermal equation of state FeO previously published by Fischer et al., *EPSL* **304**, 496 (2011) (Ref. 51). We note this procedure both in the Results section of the main text as well as in the caption for Figure 4. To theoretically estimate the equation of state (pressure vs. value at given temperature) would require the incorporation of vibrational degrees of freedom, in addition to the electronic contribution we focus on at present. This is in principle possible to do, but such challenging task is beyond the scope of current project.

Reviewer's Comments: 4) *The comparison to previous theoretical work is essentially absent. As far as I can say Ref. 50 does not show the orbital-selective QC regime. If my observation is correct, this represents a substantial difference to the previous DFT+DMFT results and should be pointed out explicitly. What is the possible origin of this discrepancy?*

Our reply: Actually, in Ref. 50, Fig. 2b, Leonov et al. found some signals related to an orbital selective regime around the Mott point, but they did not study in detail its relationship to quantum criticality. For example, at $V = V_0$ (ambient pressure) FeO is a Mott insulator, and the Fe- t_{2g} orbital has smaller gap relative to the Fe- e_g orbital, as we found as well. Upon moderate compression, at $V = 0.82V_0$, Ref. 50 finds that the gap of the Fe- t_{2g} orbital is almost closed, presenting a 'V'-shape pseudogap. Then under larger compression at $V = 0.64V_0$, they find quasi-particle peaks in both Fe- e_g and Fe- t_{2g} orbitals, as we have found in the correlated metal regime. Had these authors performed additional calculations between $V = 0.82V_0$ and $0.64V_0$, they would have seen the coexistence of the quasi-particle peaks of Fe- t_{2g} orbitals and the remaining gaps of Fe- e_g orbitals, which directly indicate the orbital selectivity. In short, while most of their results display trends consistent with our findings, the lack of sufficiently detailed data they obtained made it impossible to extract the full picture. Specifically, they presented data for only one temperature (1160 K), and only a few values of compression (volume). This clearly is insufficient to map the quantum critical region, which we find to significantly broaden with increasing temperature, and to display

characteristic pressure (volume) dependence.

[FIGURE REDACTED]

FIG. 1: The DOS of FeO at 1160 K under compression, cited from Fig.2b of PRB 101, 245144 (2020), previously Ref. 50.

Reviewer's Comments: *Is the orbital-selective QC 'phase' a realization of the state observed in 2-orbital model of PRL 99, 236404 (2007)?*

FIG. 2: The phase diagram of FeO, as a function of volume compression and temperature, cited from Fig.1 of our manuscript.

Our reply: The 2007 PRL paper by Costi and Liebsch is one of many early examples of orbital-selective Mott transitions identified using simplified model Hamiltonian, which is not of direct relevance to our realistic modeling of FeO. In addition, this paper presents results only at $T=0$ and particle-hole symmetric point, therefore does not allow them to identify or examine the QC region at finite temperature, which is the main topic of our study.

Reviewer's Comments: *5) In the methods description the author that 'The correlations treated by both the DFT and eDMFT are subtracted exactly [82]'. I find the word exactly misleading. Even in Ref. 82 (written by one of the present authors) the author used 'exactly' in quotes and the exactness is not generally accepted. I find some like*

'double counting correction of Ref. 82' a more accurate description.

Our reply: We thank the Reviewer for pointing out that our description of the methods we used needs to be improved. Our method uses a double-counting scheme that is known to be exact *within* the specific LDA-DFT scheme we utilize. To make this clear, we modified the wording of this sentence to read: "...We use a double-counting scheme [Ref. 84] which is known to be exact within the LDA exchange and correlation functional...".

Reviewer's Comments: *6) Work lacks many useful details, which should be provided at least in SM. i) Form of the interaction. Ref. 50 used density-density approximation. Slater-Kanamori or full-Coulomb interaction assuming spherical harmonic orbitals are other common choices, however, DMFT studies with less symmetric orbitals were also published. Therefore specific form should be given.*

Our reply: We thank the Reviewer for noting that the description of our method needs to be made more precise, as we have done in the revised version of the manuscript. The first section of the supplementary material now contains details of the method, such as the choice of the localized orbitals, the form of the Coulomb repulsion, the choice of the double-counting, etc.

Reviewer's Comments:*ii) Real-frequency self-energies for the presented spectra would be very helpful.*

Our reply: We thank the Reviewer for pointing out that it would be useful to add some examples of how the self-energy evolve upon compression, in order to better illustrate our findings. We agree, and therefore we added the following self-energy plots to Fig. S3 and Fig. S5 in the Supplementary Materials, illustrating the respective correlated metal and Mott insulator states of FeO.

FIG. 3: Self-energies reflecting the DOS data for $\Delta v = 0.25$ and $T = \{300, 1000, 2000\}\text{K}$ in Fig S1. The left/right column contains the real/imaginary part of the self-energy for both the eg and t2g orbitals (top and bottom rows respectively). In the top two panels, the pole generating the eg Mott gap is shown to persist for all $300\text{K} \leq T \leq 2000\text{K}$. For the t2g channel, the bottom right panel shows the zero-frequency scattering rate (i.e. magnitude of the self-energy's imaginary part at $\omega = 0$) growing with T as the quasiparticle succumbs to the heat and decoheres (thermal broadening and death of spectral peak near the Fermi energy).

FIG. 4: Self energies reflecting the DOS data for $\Delta v = 0$ and $T = 300\text{K}$ in Fig S3. The eg and t2g spectra are both gapped in this case, and the associated poles in the self-energy are shown for both orbitals.

Reviewer's Comments: *iii) Energy vs volume curves should be accessible with the present data and are needed for volume \rightarrow pressure transformation.*

Our reply: As stated above, in constructing the plots displaying the pressure dependence of various quantities, we relied upon an experimentally determined equation of state from the well known paper by Fisher et al. EPSL

304,496 (2011).

Reviewer's Comments: *iv) Formulas. Besides the e-e interaction, the authors should provide definition of M and occupation numbers (describe projections on correlated orbitals), as well as the formula to compute dc conductivity.*

Our reply: We thank the Reviewer for pointing out that a more explicit definition of the instantaneous local magnetic moment M , the orbital occupancy and electrical conductivity should be given. We have added all that information in the second section of the Supplementary Material.

REVIEWERS' COMMENTS

Reviewer #1 (Remarks to the Author):

The results presented in the paper convincingly indicate intermediate values of conductivity and relate them to a quantum critical regime above the Mott critical transition. Whereas some hints for the quantum criticality were noticed in previous work (as stressed by second referee), the earlier results were insufficient for a global picture of the state of the compound.

The relatively weak temperature and pressure dependence of resistivity could not be guessed without a broad set of calculations presented in this work and convincing labeling of the FeO under those conditions as "a quantum critical" would not be possible.

I believe that the earlier work discussed extensively by second referee is important and should be given also additional clearer credit in the manuscript, but those findings do not compromise the novelty of the main result here.

Concerning technical aspects, I find the response of the authors satisfactory and the additional information provided in resubmitted text valuable.

Concerning the term "quantum criticality" I agree with the third referee that the authors should explain the terminology clearer. Namely, couple of sentences describing the model work Terletska et al. and later associated work and discussion of how that work relates to the standard pictures of quantum criticality would be of value for the broad audience reader.

I recommend the paper for publication provided these aspects are addressed well.

Reviewer #2 (Remarks to the Author):

The authors have addressed my main concerns, and I would like to recommend it for publication as is.

A. Reviewer #1 (Remarks to the Author):

Reviewer's Comments: *The results presented in the paper convincingly indicate intermediate values of conductivity and relate them to a quantum critical regime above the Mott critical transition. Whereas some hints for the quantum criticality were noticed in previous work (as stressed by second referee), the earlier results were insufficient for a global picture of the state of the compound. The relatively weak temperature and pressure dependence of resistivity could not be guessed without a broad set of calculations presented in this work and convincing labeling of the FeO under those conditions as "a quantum critical" would not be possible... Concerning technical aspects, I find the response of the authors satisfactory and the additional information provided in resubmitted text valuable... I recommend the paper for publication provided these aspects are addressed well.*

Our reply: We thank the Reviewer for recognizing the novelty and importance of our work, for finding our results as well as our technical explanations "convincing" and "satisfactory", and for recommending publication after his comments are being addressed. We fully agree with the Reviewer's comments, which we address below.

Reviewer's Comments: *I believe that the earlier work discussed extensively by second referee is important and should be given also additional clearer credit in the manuscript, but those findings do not compromise the novelty of the main result here.*

Our reply: We thank the Reviewer for recognizing that earlier results do not compromise the novelty of our results. We agree that additional clearer credit should be given to some of the previous works, as we have done in the revised version of our manuscript.

Reviewer's Comments: *Concerning the term "quantum criticality" I agree with the third referee that the authors should explain the terminology clearer. Namely, couple of sentences describing the model work Terletska et al. and later associated work and discussion of how that work relates to the standard pictures of quantum criticality would be of value for the broad audience reader.*

Our reply: We agree with the Reviewer that the concept of "quantum criticality" should be described more precisely, with a better explanation on how it relates to previous work in the context of Mott systems, as well as its manifestations in other physical situations. To make this clear, we added a new section to Supplementary Materials., where the relevant issues are explained in detail.

B. Reviewer #2 (Remarks to the Author):

Reviewer's Comments: *The authors have addressed my main concerns, and I would like to recommend it for publication as is.*

Our reply: We thank the Reviewer for indicating that we successfully addressed their main concerns, and for recommending our paper for publication.